

# An improved low power measurement of ambient NO₂ and O₃ combining electrochemical sensor clusters and machine learning

Kate R. Smith[1], Peter M. Edwards[1], Peter D. Ivatt[1], James D. Lee[1,2], Freya Squires[1], Chengliang Dai[1], Richard E. Peltier[3], Mat J. Evans[1,2], Alastair C Lewis[1,2]

[1]Wolfson Atmospheric Chemistry Laboratories, University of York, York, YO10 5DD, United Kingdom
[2]National Centre for Atmospheric Science, University of York, York YO10 5DD, United Kingdom
[3]Environmental Health Science, University of Massachusetts, 686 North Pleasant Street Amherst, MA 01003, USA
*Correspondence to*: Peter M. Edwards (pete.edwards@york.ac.uk)

**Abstract.** Low cost sensors (LCS) are an appealing solution to the problem of spatial resolution in air quality measurement, but they currently do not have the same analytical performance as regulatory reference methods. Individual sensors can be susceptible to analytical cross interferences, have random signal variability and experience drift over short, medium and long timescales. To overcome some of the performance limitations of individual sensors we use a clustering approach using the instantaneous median signal from six identical electrochemical sensors to minimise the randomised drifts and inter-sensor differences. We report here a low power analytical device (< 200 W) that comprises of clusters of sensors for NO₂, Oₓ, CO and total VOC, and that measures supporting parameters such as water vapour and temperature. This was tested in the field against reference monitors, collecting ambient air pollution data in Beijing, China. Comparisons were made of NO₂ and Oₓ clustered sensor data against reference methods for calibrations derived from factory settings, in-field simple linear regression (SLR) and then against three machine learning (ML) algorithms. The parametric supervised ML algorithms boosted regression trees (BRT) and boosted linear regression (BLR) and the non-parametric technique Gaussian Process (GP) used all available sensor data to improve the measurement estimate of NO₂ and Oₓ. In all cases ML produced an observational value that was closer to reference measurements than SLR alone. In combination, sensor clustering and ML generated sensor data of a quality that was close to that of regulatory measurements (using the RSME metric) yet retained a very substantial cost and power advantage.

## 1 Introduction

Low cost sensors (LCS) are an attractive prospect for use in complex urban environments where more atmospheric measurements are required to build up a better resolved map of highly heterogeneous pollution patterns. There are numerous reports of low-cost, low-powered sensors commercially available for most of the criteria pollutants. Air pollution measurement has been historically a heavily regulated analytical environment. Many countries have extensive programmes of air quality measurement, and measurements often site within a legal framework with prescribed methods of measurement. Air quality monitoring stations use relatively power intensive equipment, have a high start-up cost and require skilled personnel for calibration and maintenance. A consequence is that, even in wealthy countries, observations are sparse with sites often located





1–10 km$^2$ apart (McKercher et al., 2017). Pollutants often exhibit steep spatial concentration gradients over short distances (Broday et al., 2017) and limited measurement locations mean hotspots are often missed (Mead et al., 2013).

LCS provide an opportunity to increase the density of atmospheric measurements and reduce the uncertainty that arises when

interpolating between current reference monitors. This has many uses, most notably allowing better validation of atmospheric models (Broday et al., 2017). The lower power and size associated with LCS, along with high frequency measurements, makes them an attractive prospect for mobile use and for personal exposure assessment (Williams et al., 2013). Many low-cost sensors are commercially available, either as stand-alone sensors or as multisensory platforms (Caron et al., 2016),(Jiao et al., 2016) (for example, AQMesh (Broday et al., 2017)). There has been a rapid expansion in the number of publications evaluating such

devices recently. Single devices containing sensors for the measurement of criteria pollutants such as CO, NO$_2$, total VOC and O$_3$ cost a fraction of the price (sensor box approx. cost: £5k) of establishing an equivalent measurement site with reference instruments (Mead et al., 2013) (£200k). Perhaps more importantly sensors can be placed in locations where power is limited or can only be generated through solar resources. The operating costs of low power devices also are a very attractive feature.

However, the literature contains many examples of where LCS approaches can suffer from relatively poor analytical performance, when compared against reference instruments. Whilst such a comparison is perhaps not always appropriate to make in such a highly regulated field of measurement, the benchmark test of any new analytical device will be against the regulatory reference. Significant uncertainty in measurements is introduced because individual sensors each have a unique response to simple environmental conditions such as humidity and temperature (Smith et al., 2017),(Moltchanov et al., 2015).

This can lead to a relatively high degree of inter-sensor variability and response drift (Lewis et al., 2016),(Spinelle et al., 2017) over durations as short as a few hours (Jiao et al., 2016),(Masson et al., 2015), rendering in-laboratory calibrations (where the interfering variables are controlled or non-existent) ineffective (Smith et al., 2017). Electrochemical (EC) sensors can display some chemical cross-interferences with other pollutants that are likely to be present (Mead et al., 2013),(Lewis et al., 2016),(Masson et al., 2015), and accounting for these can be difficult when the relative concentration ratios of the target

measurand and interferences change. Metal oxide sensors often lack selectivity and provide only a rough bulk measure of a particular pollutant class such as VOCs, and the responses generated can depend on the chemical of the mixture presented to the sensor.

Although some LCS vendors supply a factory calibration with their sensors, these are not always applicable in the real-world,

where ambient conditions are substantially different to the calibration conditions in the factory. Previous studies have shown that sensors co-located with reference instruments can be used to reproduce typical pollution patterns (Jiao et al., 2016),(Mead et al., 2013) but there is a significant challenge when attempting to calculate absolute pollutant concentrations with a single deployed sensor device. Recent efforts using multivariate regression models (Zampolli et al., 2004) and pattern recognition analysis (Jiao et al., 2016) have characterised these responses to the environmental conditions and provided insight into



processes that generate the sensor signal (Zampolli et al., 2004),(Hong et al., 1996). Thus far, there are no agreed standard calibration or correction procedures for sensor data, or indeed what data standards low cost sensor data should work towards. If regulatory reference methods are taken as the benchmark, the implication with current single sensors would be very frequent calibration, possibly hourly or daily. Previous work shows that clustering sensors and using the median sensor signal of the

cluster can help minimise some of the effect of medium-term noise and limit the effects of inter-sensor variability (Smith et al., 2017). This practice was adopted here during the building and development of a multi-sensor instrument deployed alongside reference instruments.

## 2 Experimental

### 2.1 Analytical description of the instrument

A range of different sensors were mounted into sealed flow-cells such that the sensing element of each was exposed to a continually flowing sample of air. The flow cells were in turn installed inside in a 4U aluminium box (177 mm H x 460 mm D x 483 mm W), which had a metal partition to keep the sensors shielded from electrical interference from the pumps and power supplies (Fig. 1). The number of sensors and their type are shown in Table 1.

Two microcontrollers (Arduino Uno) were used to collect the data from the sensors. Each Arduino recorded 3 Hz data from 25 sensors, and this was then averaged to 2 seconds and sent to a Latte Panda mini-computer for formatting and storage. Two KNF pumps drew ambient air through a sample line at atmospheric pressure over the sensors at a constant rate (c.a. 4 L min$^{-1}$). Two fans were installed on the box panels to pull air through the box in an attempt to reduce instrument overheating. The power supplies were selected for their low electrical noise, and Adafruit ADS1115 16-Bit ADC boards further minimised

this issue. A schematic of the instrument is shown in Fig. 1. The overall power budget of the device when operating was approximately 52 W, with a breakdown of components as follows: 18 x EC sensors: 9 W, 32 x MOS sensors and internal heaters: 9.4 W, 2 x RH/temp sensors: 0.01 W, 2 x diaphragm pumps: 16.8 W, 2 x fans: 2.8 W, 2 x Arduino Uno's: 0.58 W, Latte Panda micro-computer: 10 W, 3 x power supplies: 3 W.

We note that this type of approach differs from the majority of LCS air quality instruments described in the literature and that are commercially available. In most cases the emphasis in LCS design has been minimising cost and size. Clearly an instrument that contains >40 individual sensors is not optimised with cost or size as its main design goals. Instead, we have focused on data reliability as well as the advantages associated with electrical power consumption compared against a suite of traditional reference instruments.


Figure 2 summarises in simple terms how device costs and power consumption compare between a single sensor device, a six-sensor clustered approach and a reference instrument, using the example of ozone.  The clustered approach, whilst more expensive than a single sensor, retains a very substantial power advantage over the reference creating potential for deployment



in remote or off-grid locations, or in developing countries where electrical supplies can be both costly and unreliable. The next key question therefore is whether a more complex and expensive clustered sensor instrument can meet similar data quality as reference instruments, and therefore offer a direct alternative, but with lower power and operational costs.

## 2.2 Sensor test deployment in Beijing

The multi-sensor instrument described in section 2.1 was deployed alongside research-grade reference instruments in Beijing, China during a large air quality experiment between 29[th] May and 26[th] June 2017. Beijing has well documented issues around air quality (Zhang et al., 2016) meaning concentrations of pollutants were anticipated to be elevated and to show a large dynamic range. Beijing also experiences warm, humid summers (Chan and Yao, 2008); during the deployment reported here air temperature fluctuated between $15.6 - 41.2$ °C and absolute humidity ranged between $3.82 - 17.83$ g m$^{-3}$. In combination
these conditions provide a robust and wide-ranging test of instrument performance.

Both sensors and reference instruments were located at the Institute of Atmospheric Physics (IAP) site (latitude 39.978, longitude 116.387), which is situated to the north of central Beijing. All instruments were housed in converted sea container laboratories for this study. Reference instruments for $NO_2$ and $O_X$ EC were co-located and sampled from the same 3 m high inlet, with sample bypass flow provided by a common diaphragm pump. The $NO_2$ reference measurement was by cavity
attenuated phase shift (CAPS) spectroscopy (Teledyne T500U, Teledyne, California), with a 100 ppb $NO_2$ in $N_2$ calibration source. The $NO_2$ reference measurements had 5% uncertainty and 0.1 ppbv precision. $O_3$ reference was measured at 1-minute averages by a Thermo Environmental UV absorption photometer (TEI49i), traceable for calibration to the UK National Physical Laboratory primary ozone standard with an uncertainty of 2 %, and a precision of 1 ppb.

## 2.3 Data analysis approaches

The median voltage signal from of each of the sensor clusters was calculated automatically by the built-in computing device and software, and then that value converted to concentration units using four different numerical techniques: i) simple linear regression (SLR), ii) boosted regression trees (BRT), iii) boosted linear regression (BLR) and iv) Gaussian Process (GP). Machine learning techniques (methods ii-iv) are powerful tools for identifying relationships between variables and have been shown to support improved concentration estimates that correct interferences in low cost sensors(Geron, 2017),(Zimmerman
et al., 2017),(Lin et al., 2018),(Esposito et al., 2016),(De Vito et al., 2009).

The full dataset from all sensors (chemical and environmental) was used in the ML algorithms with a subset of the time-series (2[nd] June – 8[th] June 2017) treated as training data. Following training, the ML algorithms were then applied to the testing data set (8[th] June – 26[th] June 2017), outputting a corrected concentration value. The median of each sensor cluster of CO, $NO_2$, $O_3$,
VOC, plus humidity and temperature were used by the three different ML algorithms to determine the viability and relative performance of supervised, self-optimisation techniques as a method for correcting for cross interferences. Examples of both





parametric (boosted linear regression, BRL and boosted regression trees, BRT) and non-parametric (Gaussian Process, GP) techniques were assessed. BRT was chosen as a numerical method since it provides diagnostics about how the decision trees are constructed, essentially identifying which sensor signals are used in the calculation (Chen and Guestrin, 2016),(Geron, 2017). The results can then be compared to known relationships from previous laboratory studies and ensuring that the prediction is in large part a measurement rather than a model value. Gaussian Process (GP) was used because of its proven ability to handle noisy data and it ability to provide the estimations of uncertainty for each data point in the testing data (Geron, 2017),(Rasmussen and Williams, 2006).

## 3. Results and Discussion

### 3.1 How clustering improves performance

Previous laboratory studies (Smith et al., 2017) have shown that clustering sensors was one potential technical approach to reducing effects of hour to day drift in individual sensor response and limited the effects of inter-sensor manufacturing variability. The median sensor signal was shown to be a more reliable predictor of the true pollutant value (versus the mean) and the effect of deteriorating or highly variable sensors was minimised. This approach has been extended here to field observations and to a wider range of different chemical species. The EC sensors output two voltages; one from the working electrode (WE) and one from the auxiliary electrode (AE). The standard calibration procedure subtracts the effect of the auxiliary electrode from the working electrode (the electrode exposed to the ambient air and oxidising compounds) effectively helping to correct for some of the temperature and humidity effects. The manufacturer supplies individual conversion factors and equations for each sensor and these were applied to each sensor prior to use within the cluster. Each sensor within a cluster was initially normalised to give a common voltage output.

We use the raw sensor voltages and the manufacturers calibration values to gain an initial concentration. One method of determining the improvement in the concentration estimated by the sensors is to compare the range of slopes obtained against reference instrument for a range of different numbers of sensors. This is shown for the first time for an electrochemical $NO_2$ sensor in Fig. 3. As the number of sensors in a cluster is increased, the observed range of values for the unique permutations of the groups narrows considerably, greatly improving measurement precision. The slope does not however converge on 1:1 since there is a difference in the factory calibration of the sensors compared to the reference instrument. The cluster versus reference comparison using simple factory calibration can be seen in Fig. 4a.

### 3.2 Simple Linear Regression

The first data calibration approach used was simple linear regression (SLR), applied to calibrate the median sensor signal using the reference instrument concentration from the first five days of the experiment (the training period). The sensor concentrations were corrected using linear parameters from training period calibration and subsequent sensor performance was



assessed by comparing against the co-located reference instrument. Once trained, calibration factors based on SLR were then unchanged for the remainder of the experiment.

The different pollutant clusters showed variable performance against their respective reference over the 21 days. We use here root mean squared error (RMSE) as a metric to evaluate the performance of various clusters and different data calibration

approaches. We also calculate the RMSE between two approximately co-located $NO_2$ reference grade instruments (4.3 ppb) during the same field deployment to quantify what might be considered the 'optimum comparison' that could be expected between the sensors and the reference approach. During the campaign a localised source of $NO/NO_2$ was emitted into the vicinity downwind of the second $NO_2$ CAPS instrument, and hence not observed by it. For a fair comparison of the two $NO_2$ reference measurements the data between the 10th and 14th June, when the $NO/NO_2$ emission occurred, was removed.

Unfortunately, there was not a co-located CO reference instrument or multiple co-located reference observations of $O_3$ available for this study. The CO sensor median was still included with the total VOC median, RH and temperature in the sensor variables for training and testing the ML algorithms, but we were unable to make a comparison.

Applying SLR, the $NO_2$ sensor cluster gave a root mean squared error (RMSE) of 10.42 ppb and RMSE = 10.44 ppb for the $O_X$ cluster median signal with the sum of the $NO_2 + O_3$ reference measurements. The ambient $NO_2$ concentrations varied over

a wide range from below 2 ppb to in excess of 200 ppb and the clustered $NO_2$ package performed well at capturing this range of observed concentrations, but with substantial discrepancies between the median $NO_2$ EC sensor and the $NO_2$ CAPS reference instrument when the reference $NO_2$ concentrations were below 10 ppb. This finding fits well with previous work that shows the impact of cross-sensitivities on EC sensors is most important at low target compound concentrations (Lewis et al., 2016). The Alphasense OX-B431 sensors detected both $O_3$ and $NO_2$. They respond proportionately, but independently to

concentrations of $O_3$ and $NO_2$, hence the $O_X$ EC were calibrated with and compared to the sum of the $O_3$ and $NO_2$ reference measurements. The median value from the $O_X$ cluster showed the best correlation with the respective reference measurements ($O_X$ $R^2 = 0.95$, $NO_2$ $R^2 = 0.86$).

**3.3 Using machine learning (ML) algorithms to calibrate the median sensor cluster**

Each ML algorithm was trained and then tested using the same 1-minute average sensor data as the SLR in section 3.2, split into the same training and testing sets each time. The training data was the first 8490 data points of the measurement period, and the testing set the remaining 25956 data points. For BRT and BLR the python XGBoost implementation was used to train, cross validate and test the models. This scalable learning system is open source, computationally efficient, and has performed well on other platforms (Rasmussen and Williams, 2006). Both BRT and BLR have different hyperparameters that allow the

ML algorithm to be tuned so that the algorithm can detect trends within the data, without overfitting. Hyperparameters, such as the learning step can be increased or decreased to allow a good fit to the training data, and to optimise the performance of the algorithm (Geron, 2017). To tune the ML algorithm hyperparameters a five-fold cross validation of the training set was used to build the classification models, with a randomisation seed of 42 each time. The seed randomises the data, so it does not matter the value of the seed, just that it is consistent for the cross validation. During the cross validation process, the



algorithm trained on one-fold of the training data set and made a prediction based on these learnt relationships over the other four folds to test out the associated rules it has found. The hyperparameters were decided by minimising the mean absolute error (MAE) between the predicted folds and the training label (Shi et al., 2017). Once decided, these hyperparameters were fixed and the algorithm then tested on data that it has not yet seen, i.e. the testing data set.

BRT uses gradient boosted regression trees to integrate large numbers of decision trees, and this improves the overall performance of the trees (Rasmussen and Williams, 2006). Through a process where many decision trees are working on the training data set the algorithm generates a set of rules by which the training data is linked to the training label (Shi et al., 2017). By discarding trees that do not have much impact on the MAE, the algorithm is more efficient at determining the relationships

between variables. The nature of decision trees means BRT is not limited to identifying linear functions, unlike BLR. During the same cross validation process as described for BRT, BLR identifies the linear relationships between the sensor variables and uses these correlations to predict the compound response during the testing period. BLR is simpler than BRT but works well when there are multiple linear trends between variables. Gaussian Process (GP) uses the Gaussian distribution over functions and can be a powerful tool for regression and prediction purposes (Rasmussen and Williams, 2006). It is a flexible

model which generalises the Gaussian distribution of the functions that make up the properties of each variables function (Rasmussen and Williams, 2006). GP can be used as a supervised learning technique once suitable properties for the covariance functions (kernels) are found, then a GP model can be created and interpreted. For this study there were two kernels used to train and predict the sensor data. These were Matern32 (k1) and Linear (k2) functions. They were added together (k1 + k2) to enable both linear (k2) and non-linear (k1) relationships between the variables to be detected, as it was observed in the

laboratory that the relationships between the variables could be either (Lewis et al., 2016),(Smith et al., 2017). The hyperparameters were then self-optimised using the training data by the open-sourced python package running the algorithm, GPy. The GP, BRT and BLR predicted responses were then compared to the reference data over the testing period, and a RMSE calculated to investigate how well the ML algorithm performed.

**3.4 Sensor cluster data with ML processing – NO₂ cluster**

Figure 4 shows the predicted $NO_2$ time series using the median cluster value and the three ML calibrations compared with the reference measurement. The median sensor with individual factory corrections (Fig. 4a) clearly detects the major trend in $NO_2$ concentration, but often under predicts at times when the $NO_2$ concentration is low. At higher concentrations the median sensor overpredicts the $NO_2$ signal, leading to a RMSE of 86.7 ppb.

**3.4.1 Gaussian process (GP)**

The GP ML algorithm predicted the $NO_2$ concentration with a RMSE of 5.2 ppb compared to the reference measurement, the lowest for all the different ML techniques. The Matern32 kernel is adept at capturing the more typical (sub 50 ppb) $NO_2$ concentrations, due to its ability to model cross-sensitivities on the sensor signals but struggled to extrapolate to highest concentrations. One advantage of using GP to predict compound concentrations is that an uncertainty on the predicted values





is also calculated. This uncertainty is shown in Fig. 4b (light yellow shading), as $\pm$ 2 standard deviations on the predicted data points. It is clear that there are periods when there is more uncertainty in the prediction. There are four main periods where the GP prediction appeared low, and the uncertainty was high: 1500H 8[th] June, 1700H 9[th] June, 1400H 15[th] June and 1400H 16[th] June. These over-extrapolated data points all occurred when the temperature reached +40 °C and exceeded the maximum

temperature recorded during the training period (35.8 °C), coinciding with the $NO_2$ concentration and RH were low (Fig. 4e). Machine learning techniques all have difficulty making predictions when the testing and training data sets cover different variable space, but the calculation of a prediction uncertainty which takes this into account highlights when this could potentially be an issue and could be used to inform calibration strategies.

**3.4.2 Boosted Regression Trees (BRT)**

The BRT prediction (Fig. 4c) was very good during periods when the test data did not exceed concentrations of $NO_2$ seen in the training data (~79 ppb). However, the classification nature of the BRT algorithm means it is incapable of extrapolation, so the prediction cannot capture the high concentrations of $NO_2$ that were observed between the 10[th] – 14[th] June (the $NO_2$ CAPS instrument recorded a maximum $NO_2$ concentration of 222.2 ppb during the testing period). Between this time a localised

source of $NO/NO_2$ was emitted. Overall, the RMSE between the BRT $NO_2$ prediction and the $NO_2$ CAPS reference measurement was 7.2 ppb, an improvement on SLR (10.4 ppb) of ~30% despite its inability to capture $NO_2$ concentrations outside of those experienced during the training data period. This improvement for the lower concentrations of $NO_2$, is due to the BRT model's ability to better correct for some cross sensitivities on the sensor signals, such as the effect of humidity. With the dates omitted for the localised source of $NO/NO_2$ (described in section 3.2) the RMSE for BRT prediction was 6.1 ppb,

showing that the BRT prediction does well at capturing the trends in $NO_2$ when extrapolation is not required.

The BRT algorithm outputs a gain feature called gain, which can be used to identify how much each variable contributes to the predicted sensor response and these are shown in Fig. 5.  The median $NO_2$ sensor signal was (encouragingly) the largest contributor to the $NO_2$ concentration prediction, followed by data from the CO cluster and the relative humidity sensor. This is consistent with previous laboratory results, where it was observed that the $NO_2$ sensor signal had a CO interference and was

affected by changing humidity (Lewis et al., 2016).

**3.4.3 Boosted Linear Regression (BLR)**

The BLR predicted $NO_2$ concentration was comparable to the GP prediction, with a RMSE of 6.6 ppb. When the $NO/NO_2$ localised source was removed the RMSE did not change substantially (6.3 ppb) suggesting that this technique was good at extrapolating to the $NO_2$ concentrations outside the range of the training data. BLR assumes purely linear trends between

variables, meaning it does not represent non-linear relationships, but the linear nature of the relationships allows BLR to extrapolate trends beyond the ranges seen in the training data. Figure 5d shows the predicted BLR $NO_2$ signal fully capturing the maximum $NO_2$ concentrations between the 10[th] – 14[th] June. Overall, the RMSE between the BLR prediction and $NO_2$ reference measurement were slightly better than the BRT suggesting that the inter-sensor relationships were often approximately linear over the variable space observed. The similarity between the GP and BLR predictions are not surprising





given the use of the linear kernel in the GP algorithm. The BLR also over-extrapolated the predicted $NO_2$ concentration during the same periods as the GP prediction, suggesting that the linear kernel contributed substantially to the GP prediction but that the training data was not adequate to capture deviations from this linearity.

Figure 7a summarises how a progressively improved RMSE can be achieved as $NO_2$ sensors are first used in a cluster, and then the various different numerical methods applied to calibration, ultimately producing performance that is close to the reference vs reference RMSE. Figure 7a also highlights the evidence that the uncertainty in the sensor concentrations is greatly reduced if the sensors are calibrated in field (using SLR) or if ML procedures are applied. The GP prediction was the ML calibration technique that was closest to the RMSE between the two reference instruments.

### 3.5 Sensor cluster data with ML processing – $O_x$ cluster.

The data from the median $O_X$ sensor versus the $NO_2 + O_3$ reference measurements is shown in Fig. 6, along with the best performing ML data processing method. During peaks in $O_X$ concentration the factory calibrated sensor values tend to produce over estimates of the $O_X$ concentrations (e.g. maximum $O_X$ concentration observed by reference was 253 ppb, the median $O_X$

sensor 426 ppb). The best performing ML calibration technique was BRT, likely due to the training data set containing a similar range of $O_X$ concentrations to the testing data.

A summary of RMSE improvements, implemented for all methods can be found in Fig. 7b. BLR and BRT performance was near identical indicating the $O_X$ sensors have largely linear relationships governing their performance, at least over the variable space observed. The 30% of the data used to train the ML algorithms included a range of $O_X$ concentrations much more

representative of the total observation period than was the case for $NO_2$, and so only limited extrapolation beyond the training dataset was needed. The BRT algorithm gain was again used to determine the largest contributing variables to the BRT $O_X$ prediction. The median $O_X$ sensor value made the largest contribution to the BRT $O_X$ prediction (93%). The median CO sensor contributed 1.5% to the prediction.

### 3.6. A measurement vs a sensor model

ML algorithms are skilful at detecting patterns within a dataset and the work shown in this study is evidence that they can improve the performance of LCS, as measured by a reported concentration value compared to a reference. Each of the sensor predictions made by the ML algorithms could be justified by previous experience with working with similar EC sensors in the laboratory and from reported studies. For example, the predicted $NO_2$ sensor response was formed based upon decision trees

that were primarily influenced by the median $NO_2$ sensor value, then small adjustments were made to the prediction using the median CO EC and humidity data. This is reasonable based on previous laboratory experiments showing $NO_2$ sensors responding to CO and changing humidity. When using the sensors to correct cross interferences and changing meteorological conditions, the prediction is an optimised version of the sensor response that essentially calibrates for identified cross-sensitivities.



However, ML algorithms can also be used to make predictions of compounds, for example nitric oxide (NO), that are simply correlated to other air pollution variables, but that are not physically measured by a specific sensor. As an example, in this study a reference grade NO measurement was made from the same sampling line as the sensor instrument and this was used

to make a NO-prediction using BRT, based on information gathered by the other chemical sensors. From previous laboratory studies it is known that NO is a cross interference on the $NO_2$ and $O_X$ EC sensors (Lewis et al., 2016), and therefore we could expect that an NO prediction would use these two variables. However, ambient NO concentrations are closely linked to the concentrations of $NO_2$ and $O_3$ via steady state inter-conversion, and this underlying chemistry might also be identified by the algorithm and used to predict NO.

Using a BRT model and sensor cluster median values from the sensor instrument deployment, it was possible to correctly identify when the major NO peaks would occur and predict NO concentrations with a RMSE of 10.5 ppb, even though our instrument did not actually contain a NO sensor. This corresponds to a Normalised Root Mean Squared Error (NRMSE) of 0.37. For comparison, the NRMSE for the BRT $NO_2$ and $O_X$ predictions were 0.11 and 0.08 respectively, and the two $NO_2$ reference instruments gave a NRMSE of 0.06, so the NO prediction contains a high degree of uncertainty although appears to

be quite good initially. When we interrogate the decision tree model however, we find that the prediction is largely based on the chemical relationship between $NO_2$ and $O_X$, and not on any cross-sensitivities on sensor signals. In this rather extreme example it could be claimed that this NO prediction is not a measurement but a model (Hagler et al., 2018), and highlights the challenge of interpreting low cost sensor measurements that exist in something of an analytical grey area due to their reliance on complex calibration algorithms.

## 4. Conclusions

Using a combination of clustering sensors and machine learning data processing, a lower cost and relatively low power air quality instrument has made measurements of $NO_2$ and $O_X$ that were close to the RMSE of reference instruments (over the period of study). Clustering of sensors adds little to the overall power budget of an instrument but is a very easy way to

overcome individual sensor drift and irreproducibility. Further data treatments such as in-field calibration with SLR or supervised ML techniques can further optimise the sensor data. SLR was seen to improve median sensor concentrations to some degree but struggled to accurately calibrate the sensor data at the lower concentrations. ML techniques were able to further improve the sensor performance because they could correct multiple trends between the sensor variables eliminating some cross-interferences. BLR and BRT were seen to be most powerful at predicting the compound response and used

information content from other variables that was reasonable based on previous lab studies. The GP approach was advantageous in that a standard error could be calculated for each predicted data point. Therefore, this identified regions within the data where the prediction was more uncertain, for example, if the testing data significantly deviated from the variable space observed during training. BLR was the simplest technique and worked well when the functions between the sensor variables were linear, for example during the $O_X$ sensor prediction. The time required to train and run the model was reduced when using



BLR and BRT over GP. A longer period of data collection, of at least a few months to a year of sensor data, is needed to establish how long such algorithms accurately predict the reference observations. It appears that as a minimum the use of ML calibration techniques would increase the time required between physical calibrations and allow the use of sensor instruments as part of a network or to run in isolated environments, after the instrument was calibrated over as large a range of conditions

it is likely to experience as possible. Data that occurs outside the training data ranges can then be flagged and treated with a higher level of uncertainty.

## Author contributions

KS, PE, designed and developed the sensor instrument. KS, PE, PI and CD contributed to analysis of sensor data.

FS, JL and YS provided reference data. All authors contributed to the writing of the manuscript.

## Acknowledgements

AIRPRO grant NE/N007115/1, AIRPOLL grant NE/N006917/1, NCAS/NERC ACREW, Peter M. Edwards acknowledges a Marie Skłodowska-Curie individual fellowship, Kate R. Smith and Freya A. Squires acknowledge NERC SPHERES DTP

PhDs. Peter D. Ivatt acknowledges an NCAS Studentship PhD.

The authors declare that they have no competing interests.

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

The reference data can be found on the CEDA website under the Atmospheric Pollution and Human Health in a Developing Megacity (APHH) project.



**Table 1: Summary of sensors used within the instrument.**

| Measurand | Sensor type | Manufacturer | Number of sensors in each cluster | Number of clusters |
|---|---|---|---|---|
| Carbon monoxide (CO) | Electrochemical CO-B4 | Alphasense | 6 | 1 |
| Oxidising gases (O$_X$) | Electrochemical OX-B431 | Alphasense | 6 | 1 |
| Nitrogen dioxide (NO$_2$) | Electrochemical NO2-B43F | Alphasense | 6 | 1 |
| Total VOC | Metal oxide TGS2602 | Figaro | 8 | 4 |
| Temperature and humidity | Transducer (HPP809A031) | TE Connectivity | 1 | 2 |



**Figure 1: Schematic representation of the gas flow-paths and basic layout of the sensors and components within the device.**




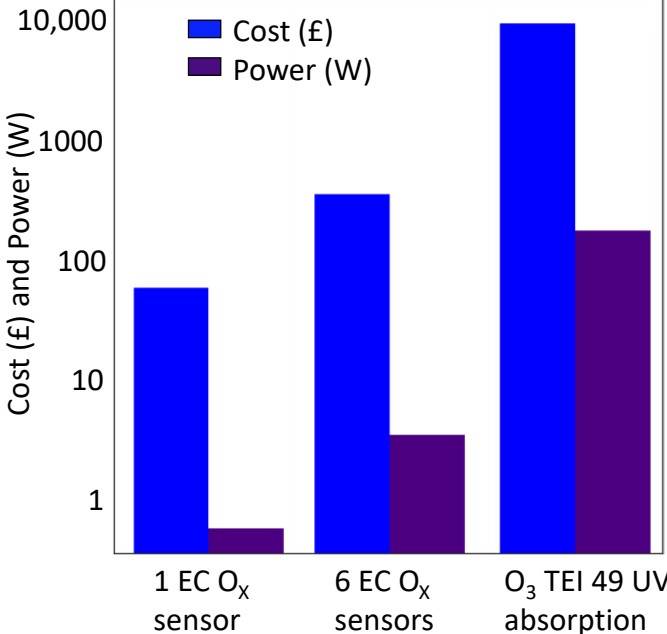

**Figure 2: Cost (blue) and power (purple) competitiveness for a single $O_X$ EC sensor device, a clustered six-sensor device and a reference UV ozone monitor.**





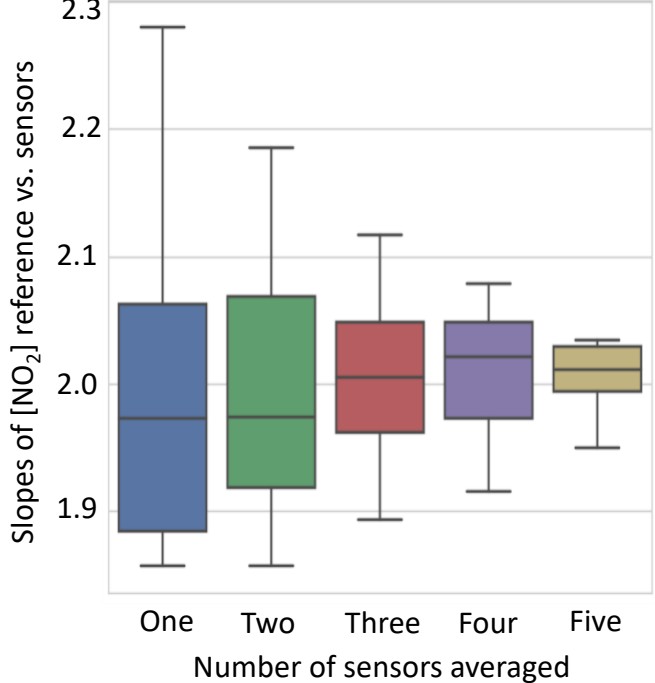

**Figure 3: Comparison of slopes of concentrations derived from clusters of NO₂ EC sensors against a reference instrument for ambient Beijing air. As the number of sensors used increases, the spread in data, as seen through the difference in slope, narrows. If data from 3 out of 6 sensors is used there are 20 possible permutations of sensors. The average signal of each was calculated, then plotted against the reference NO₂ CAPS measurements and the gradient extracted. The 20 gradients of these correlation plots (sensitivities) are then plotted in the boxplots above, with the median, 25th percentile, 75th percentile in the box and the 5th and 95th percentile on the whiskers.**




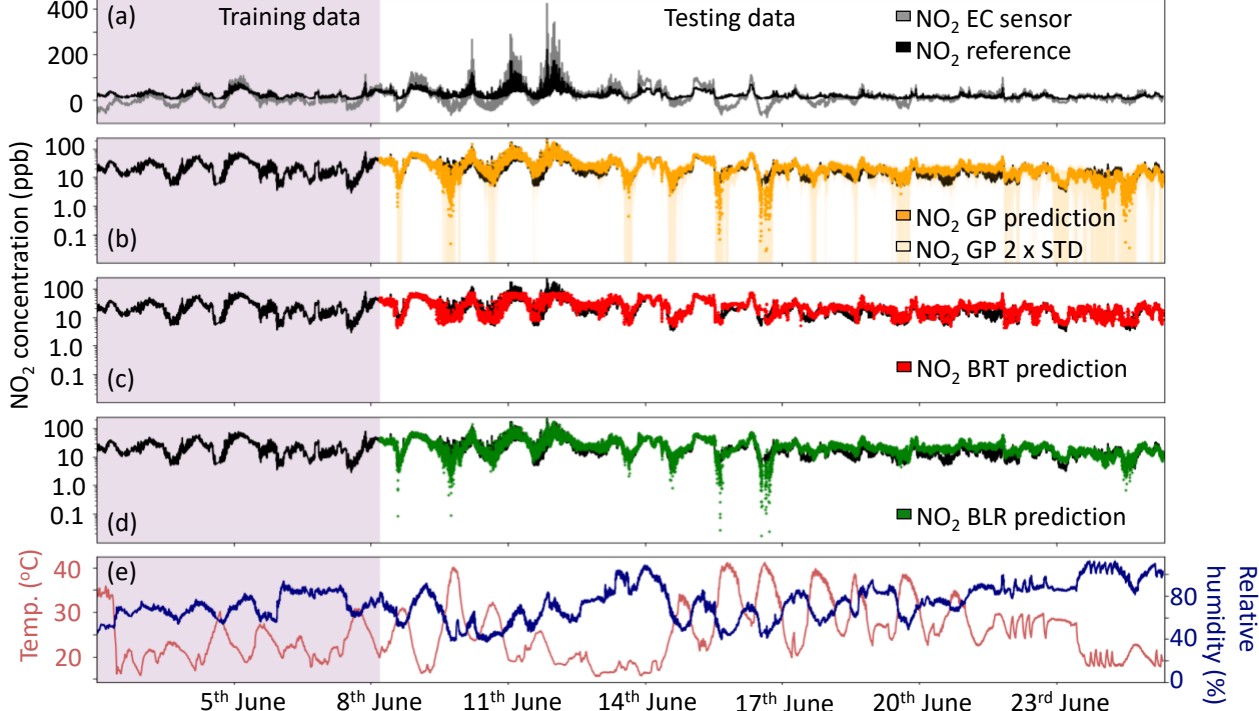

**Figure 4: a) Comparison of the median NO₂ sensor using individual factory calibrations, (b) the NO₂ GP prediction ±
2 σ, (c) NO₂ BRT prediction and (d) NO₂ BLR prediction ML techniques. The purple shaded area shows the data used
to train the ML algorithms. The black line in all subplots is the York NO₂ CAPS measurement, which was used as a**
5   **reference. Panel (e) shows the relative humidity (%) and temperature (ºC) during the sensor instrument deployment.
N.B. Panels (b), (c) and (d) are plotted with a logarithmic y-axis.**





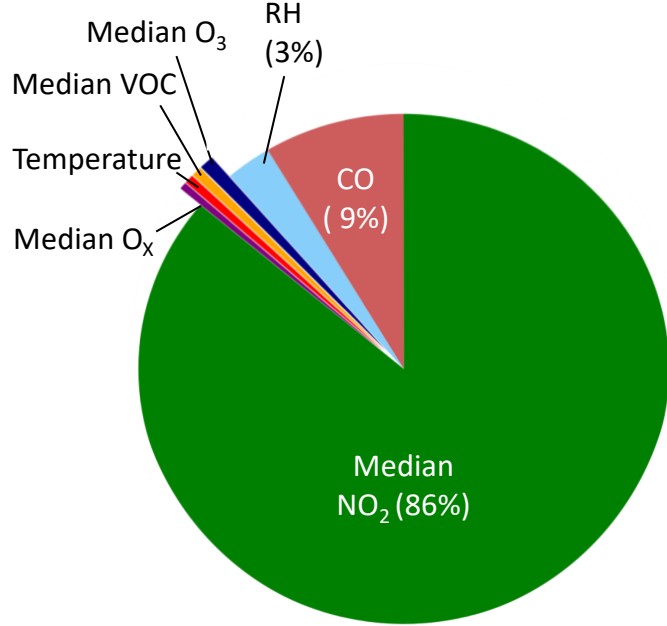

**Figure 5: Breakdown of contribution from each variable used by the BRT algorithm to predict the clustered NO$_2$ sensor concentration.**



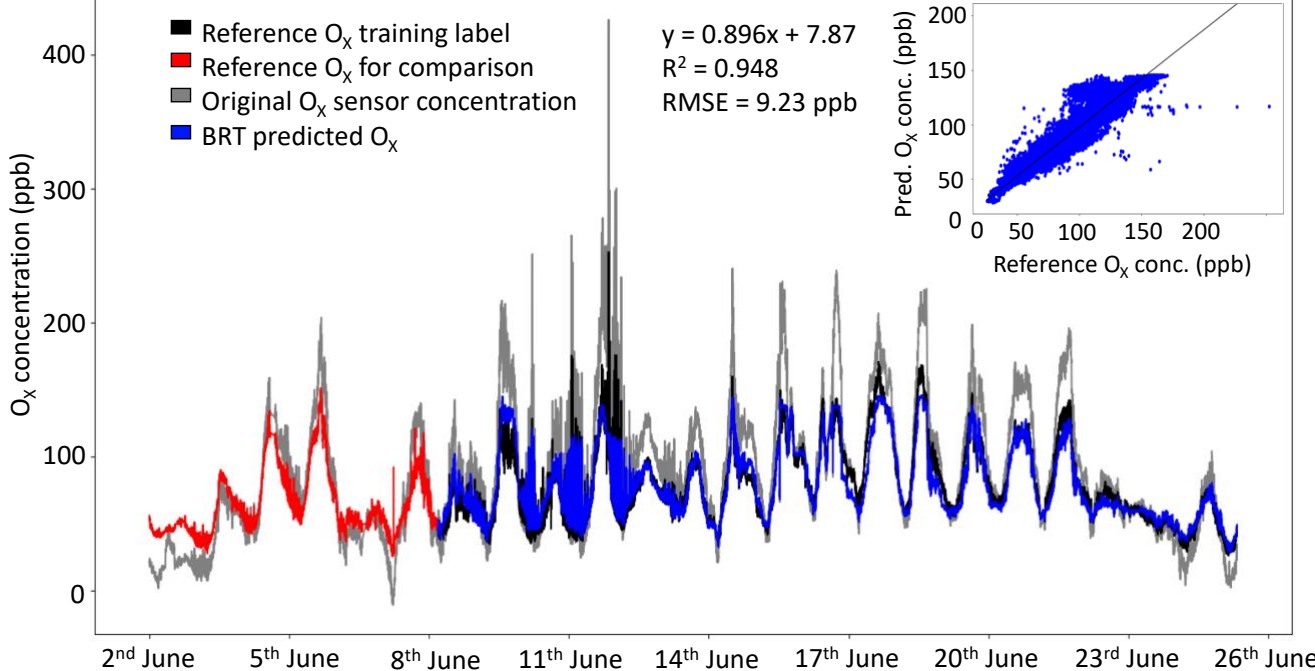

**Figure 6: Factory calibrated median sensor concentration (grey), reference O₃ + NO₂ data (black) and BRT Oₓ prediction (blue) for cluster of Ox sensors. The reference measurements that were used as the training label are displayed in red. Inset: The correlation plot for the testing dataset, comparing the reference data and the BRT predicted Oₓ sensor signal.**





**Figure 7: Comparison of the RMSE calculated for electrochemical sensor signal data treatment including individual sensors and a cluster of six using factory calibration, SLR and three ML techniques; when available, a reference versus reference RMSE is also included.  a) NO₂, b) Oₓ.**

