# Peer review of "An improved low power measurement of ambient NO2 and O3 combining electrochemical sensor clusters and machine learning"

_Atmospheric Measurement Techniques, 2018_

## Referee Comment (RC1) · Anonymous Referee #1 · 26 Nov 2018

General Comments Overall this a well-written, well-organized contribution to the low-cost sensor literature. The authors demonstrate the importance of bespoke sensor calibration using several techniques, including sensor clustering and various statistical and machine learning techniques. Sensor clustering reduces uncertainty due to inter-sensor differences and overall accuracy is improved using several statistical/machine learning techniques.

Specific Comments -How accurate is 'good enough'? I think a bit more context regarding this question would be helpful to readers.

-Overall, using SLR and ML techniques seems to be the largest source of improvement.

[Figure]

Is sensor clustering even necessary?

-Does sensor accuracy vary over the observed concentration ranges?

Technical Comments p1 l30. 'site'->situated p2 l8, l19, l21... Check reference parentheses throughout p4 l25. Also Hagan et al. AMT 2018

———————————————————

---

## Referee Comment (RC2) · Anonymous Referee #2 · 31 Dec 2018

Comments

Low cost sensors (LCS) playing an emerging role in the urban environmental monitoring with respect to the possibility of setting up a densely populated gridded network. Nevertheless, the detection limit, the stability and the real-time calibration were in general of question or with difficulty to overcome. In this study, the authors try to use the machine learning (ML) method to enhance of the data quality of LCS which is in general fit the effort of the community to improve the data quality of LCS. The paper is within the scope of AMT and I have the following comments for the authors to consider before publication.

1. The machine learning method is used to improve the data quality of the LCS. The improvement is clear but still without in-depth explanations. The scientific paper shall not be looks like simply magic. I will be convinced if the authors can provide much more examples as the authors also wrote in their conclusions. Moreover, I did see much better comparison results from LCS (the Cambridge group for the same campaign) with the CAPS instrument on NO2 and other parameters like O3, CO, etc. So, I wonder if the results presented in this paper can be improved further.

2. Sect. 3.2: during the training period, what kind of regression method is used to calibrate the sensors? According to Cantrell, 2008(Cantrell, C. A.: Technical Note: Review of methods for linear least-squares fitting of data and application to atmospheric chemistry problems, Atmos. Chem. Phys., 8, 5477–5487, 2008.), bivariate regression algorithm is required to retrieve the robust slope.

3. Figure 4, Panel A is with linear scale, Panel B-D is with logarithmic scale. Why the authors want to have two different scales?

4. Figure 5 is a nice way to explain the advantage from the ML method. Can the authors do the same for the other ML processing?

5. The ML corrected LCS signal still significantly smaller than those measured by the reference instruments especially for the peak values of Ox? Could the authors provide more discussions on this aspect and what could be the possible improvements on LCS or ML.

Technical comments:

In most cases, the multi-citations were not correctly implemented. For example, page 2 line 8, (Caron et al., 2016),(Jiao et al., 2016) should be (Caron et al., 2016; Jiao et al., 2016). This shall be revised throughout the paper. Figure 3 is not cited in the main text which I assume should appear somewhere in Sect. 3.2.

---

## Author Comment (AC1) · 4 Feb 2019

Specific Comments

C1 : How accurate is 'good enough'? I think a bit more context regarding this question would be helpful to readers.

C1. Author's Response Thank you to the reviewer for the advice to be clearer when describing the requirements for sensor performance before they are considered able to perform as instruments in the field. The answer to this question is very application dependant and we cannot therefore provide a definitive statement on how good is

good enough. It is for this reason that we included the comparison for NO2 measured by two identical reference grade instruments in order to provide some reference for our comparisons. In order to try and expand on this point, we have added some text to the manuscript (page 3, lines 2 - 8) describing the standards set for reference grade instrument performance as set by the EU Directive 2008/50/EC, Annex 1(EU, 2008). Conforming to these standards is an obvious target for low cost sensor measurement performance, but that is not to say that reduced accuracy observations do not hold value providing the uncertainties are quantified.

C1. Changes to manuscript Page 3, lines 2 - 8 For reference monitors in the UK, NOX, CO and O3 instruments must produce reproducible measurements for three months that are within 5% of the average for a certain concentration in the field, and results that are linear over a set range (EU, 2008). For NOX this is 0 – 2000 ppb and O3: 0-500 ppb and CO: 0 – 50 ppm to ensure that both rural and urban concentration ranges are taken into account. Although the target performance of low-cost sensors is highly application dependent, these standards do provide a benchmark for comparison and highlight the need not only for high accuracy measurements but also reproducibility over long (months) timescales. In order for low cost sensors to be used in atmospheric monitoring or research applications the uncertainty and reproducibility must be quantified across a range of likely environmental conditions.

C2. Overall, using SLR and ML techniques seems to be the largest source of improvement. Is sensor clustering even necessary?

C2. Author's response Ultimately the clustering and statistical calibration methods are performing different functions in improving sensor performance. In terms of measurement accuracy, the SLR and ML calibration algorithms provide significant improvement over simple linear regression, due to their ability to correct for the multiple cross-sensitivities on sensor signals. In contrast, the function of the clustering approach is not to improve measurement accuracy, but rather sensor reproducibility. As shown in our previous paper (Smith et al., 2017) many sensors show variability in both signal and

sensitivity over timescales of days or longer. This variability is very difficult to remove through time averaging, however the lack of correlation of this noise / drift between identical sensors means it can be addressed by instead averaging over multiple sensors. The conclusion of Smith (2017) was that clustering greatly reduces medium-term random noise in the average sensor signal, thus improving confidence in sensor signals and in theory prolonging the time requirements between calibrations. The time series in Fig 1. illustrates how sensor signals drift apart over time. The plot on the left shows all six sensor signals immediately after calibration to a reference monitor, showing a tight clustering around the median value (red). The plot on the right however shows the drift in individual sensors after a period of 16 days. The use of an average sensor reduces some of this signal variability enabling a more robust calibration to be applied, using algorithms such as SLR or Gaussian Process etc. The following text was added to the manuscript to emphasise that clustering and ML are used to target different issues. Page 4, line 26 -27.

C2. Changes to manuscript The clustering approach was used to improve sensor reproducibility as previously discussed in (Smith et al., 2017), whereas the SLR and ML techniques were applied to improve sensor accuracy by correcting for cross sensitivities.

C3. Does sensor accuracy vary over the observed concentration ranges? C3. Author's response This is an excellent question by the reviewer, and we have performed additional analysis below to investigate this. For each calibration method used in the paper, the data was 25 % of the observed reference concentration range bins. The Root Mean Squared Error (RMSE) and the Normalised Root Mean Squared Error (NRMSE) were calculated for each concentration bin and the results for NO2 and OX are summarised in the tables below. The NRMSE was calculated by dividing the RMSE between the reference observations and the sensor values by the mean reference concentration for the respective bin. Table 1 and 2 nicely displayed how the different analytical techniques improved the sensor performance at different concentrations. Therefore, we

decided to include Table 1 (Fig. 3 this document) and a description of the results in the manuscript summarising the NO2 RMSE and NRMSEs. The OX summary was very similar so wasn't included, but the authors are happy to include it if the editor wishes. Table 2 (Fig 4. of this document) summaries the results for the OX analysis.

C3. Changes to manuscript- added in a Table (Fig 3) to summarise these results on page 16 of the manuscript.

Text changes: Page 9, line 18 - 26 The RMSE and NRMSE was calculated after the application of SLR and ML for different reference concentration ranges to indicate where the greatest improvement of the sensor data occurred (see, Table 2). The RMSE and NRMSE (calculated by dividing the RMSE by the mean of the concentration bin) were determined between the reference NO2 observations and the sensor values for four equally spaced reference concentration bins. The ML techniques produced the greatest improvements in the concentration estimates for the lower concentrations of the target measurand where the effect of cross interreferences is more significant. The BRT and GP in particular displayed large improvements for the lower NO2 reference observations. At the higher concentrations of NO2, the ML algorithms displayed less improvement, where the conditions were outside those of the training data variable space. This was very noticeable for the BRT algorithm due to its inability to extrapolate.

Page 10 line 8 - 11. The NRMSE was calculated for 4 equally sized reference OX concentration bins for each analytical method used, in a similar manner to Table 2 for NO2. The NRMSE improved for SLR and the ML algorithms across all concentration ranges, with BLR and BRT optimal for reducing the error estimate the most. The error was the highest at the higher OX concentrations for BRT, which was expected due to BRTs inability to extrapolate.

C4. Technical Comments p1 l30. 'site'->situated C4. Authors response The wording has been changed on page 1, line 32.

C5 . p2 l8, l19, l21... Check reference parentheses throughout p4 l25. C5. Author's response Removed the extra brackets between multiple references and inserted a semi-colon to differentiate two citations for the same reference.

C6. Also Hagan et al. AMT 2018 C6. Author's response Added the reference into the manuscript on page 5, line 3 as it was relevant to the manuscript.

References EU: Directive 2008/50/EC of the European Parliament and of the Council of 21 May 2008 on ambient air quality and cleaner air for Europe, Eur. Union, 1–62, 2008.

Smith, K., Edwards, P. M., Evans, M. J. J., Lee, J. D., Shaw, M. D., Squires, F., Wilde, S. and Lewis, A. C.: Clustering approaches that improve the reproducibility of low-cost air pollution sensors, Faraday Discuss., 00(0), 1–17, doi:10.1039/C7FD00020K, 2017.

Please also note the supplement to this comment:
https://www.atmos-meas-tech-discuss.net/amt-2018-285/amt-2018-285-AC1-supplement.pdf
* * *
a) One hour sensor time series immediately after calibration

b) One hour sensor time series 16 days after calibration

Individual $O_X$ EC sensors
Median $O_X$ sensor

Calibration

$O_X$ concentration (ppb)

Time on 8th June 2017

Time on 24th June 2017

**Fig. 1.** Six individual OX EC (blue) with the median OX EC (red), a) immediately after SLR calibration with the reference observations and b) 16 days after the calibration.

[Figure]

**Fig. 2.** Six individual NO2 EC (green) with the median NO2 EC (purple), a) immediately after SLR calibration with the reference observations and b) 16 days after the calibration.

| | NRMSE of Reference vs. $NO_2$ concentration estimate (RMSE / ppb) | | | | |
|---|---|---|---|---|---|
| Concentration range as a % of the max. conc. of reference $NO_2$ | Median | SLR | BLR | BRT | GP |
| 0 - 25 % | 1.04 (20.7) | 0.59 (11.7) | 0.32 (6.3) | 0.28 (5.6) | 0.29 (5.8) |
| 25 - 50 % | 0.69 (47.5) | 0.19 (13.3) | 0.12 (8.2) | 0.22 (15.2) | 0.11 (7.9) |
| 50 - 75 % | 0.72 (94.9) | 0.23 (30.8) | 0.26 (34.6) | 0.55 (72.5) | 0.26 (33.5) |
| 75 - 100 % | 0.85 (153.1) | 0.10 (17.4) | 0.10 (18.8) | 0.67 (120.0) | 0.10 (18.2) |

**Fig. 3.** Table 1 The NRMSE and RMSE between the NO2 reference and sensor data sets at different concentrations ranges. For each calibration method used in the paper, the data was binned into 25% of the observe

| | NRMSE of Reference vs. $O_X$ concentration estimate (RMSE / ppb) | | | | |
|---|---|---|---|---|---|
| Concentration range as a % of the max. conc. of reference $O_X$ | Median | SLR | BLR | BRT | GP |
| 0 - 25 % | 0.21 (11.0) | 0.16 (8.4) | 0.10 (5.4) | 0.12 (6.0) | 0.18 (9.2) |
| 25 - 50 % | 0.30 (26.4) | 0.12 (10.2) | 0.11 (9.4) | 0.11 (9.7) | 0.14 (12.4) |
| 50 - 75 % | 0.36 (50.4) | 0.12 (16.3) | 0.12 (16.1) | 0.10 (14.0) | 0.16 (22.4) |
| 75 - 100 % | 0.52 (116.1) | 0.20 (44.7) | 0.26 (58.0) | 0.49 (110.9) | 0.27 (60.6) |

**Fig. 4.** Table 2. The NRMSE (and RMSE) between the OX reference and sensor data sets at different concentration ranges.

[Figure]

---

## Author Comment (AC2) · 4 Feb 2019

General Comments Low cost sensors (LCS) playing an emerging role in the urban environmental monitoring with respect to the possibility of setting up a densely populated gridded network. Nevertheless, the detection limit, the stability and the real-time calibration were in general of question or with difficulty to overcome. In this study, the authors try to use the machine learning (ML) method to enhance of the data quality of LCS which is in general fit the effort of the community to improve the data quality of LCS. The paper is within the scope of AMT and I have the following comments for the authors to consider before publication.

[Figure]

C1. The machine learning method is used to improve the data quality of the LCS. The improvement is clear but still without in-depth explanations. The scientific paper shall not be looks like simply magic. I will be convinced if the authors can provide much more examples as the authors also wrote in their conclusions. Moreover, I did see much better comparison results from LCS (the Cambridge group for the same campaign) with the CAPS instrument on NO2 and other parameters like O3, CO, etc. So, I wonder if the results presented in this paper can be improved further.

C1. Author's response We thank the reviewer for this comment, as this is something we really wanted to avoid and have therefore added more detail in order to try and be more explicit about this. During the analysis section of this work the authors made sure that the ML techniques used provided outputs on the decisions they made that could then be compared with laboratory experiments and previous sensor studies, in order to make the methods used not seem like black boxes. This was underpinned by our choice of ML techniques; BRT was chosen because of the function to extract out the variables gain contributions, GP could produce the uncertainty for each predicted data point and the weights associated with each variable can be extracted from the BLR algorithm (see C4 with Figs. 4 and 5 of this document). The manuscript aimed to compare results from using different techniques on the same dataset and therefore, fully comprehensive explanations of different ML techniques is beyond the scope of this paper. However, we recognise that this was not made clear enough in the manuscript so have added some more detail in the text and some more citations to detailed descriptions of the techniques.

With respect, the authors have not seen any publications from the Cambridge group on this and cannot comment on unpublished work. Whilst there are many references to studies where LCS have been used successfully in the field, the scope of this manuscript relates to the improvement of low-cost sensor performance for deployment over longer periods of time and possible calibration strategies that would enable this.

C1. Changes to manuscript Gaussian process reference inserted: Gaussian process

for time series modelling, S. Roberts, M. Osborne, M. Ebden, S. Reece, N. Gibson and S. Aigrain (Roberts et al., 2013) Page 7, line 27.

XGBoost reference inserted: Greedy function approximation: a gradient boosting machine (Friedman, 2001). Page 7, line 17.

C2. Sect. 3.2: during the training period, what kind of regression method is used to calibrate the sensors? According to Cantrell, 2008 (Cantrell, C. A.: Technical Note: Review of methods for linear least-squares fitting of data and application to atmospheric chemistry problems, Atmos. Chem. Phys., 8, 5477–5487, 2008.), bivariate regression algorithm is required to retrieve the robust slope.

C2. Author's response Thanks for bringing to our attention that the SLR method described in Section 3.2 was unclear. More text has been added to better describe the linear regression process. There were four different types of analytical techniques used in turn to examine the performance of ML versus simple linear regression (SLR). In section 3.2, SLR was used to calibrate the EC sensors against their respective reference instruments.

Using NO2 as an example, linear parameters in the form of y = mx + c were determined using a linear least squares fit between the NO2 CAPS reference instrument and the median NO2 EC sensor. This linear relationship was calculated for the first five days of the deployment – the same five days that were used as the training period for the ML analysis.

Text has been added on Page 6 lines 6 – 10, to provide further detail about SLR.

The training period for the NO2 EC sensors and NO2 reference measurements was also re-analysed using bivariate regression (Ordinary Least Squares) and the resulting model was applied to the median NO2 sensors over the testing period. This produced a bivariate regression NO2 prediction, shown in green in Figure 1. The regression was performed using the Python statsmodels package.

The RMSE was calculated between the BR NO2 prediction and the NO2 reference measurements in the testing period and was found to be 14.6 ppb. The bivariate regression prediction therefore contains more error than the simple linear regression in the manuscript (10.42 ppb). This does not change the outcome of the paper which aims to use ML to improve the quality of the NO2 sensors by correcting for cross interferences and therefore has not been added to the manuscript. The SLR was used to show the calibration of the sensors using linear regression and essentially set a baseline for the improvements.

C2. Changes to manuscript Using the NO2 EC as an example, linear parameters in the form of y = mx + c were determined using a linear least squares fit between the NO2 CAPS reference instrument and the median NO2 EC sensor for the first five days of the sensor instrument deployment. Once trained in this manner, these linear calibration factors based on SLR were used to calibrate the median NO2 sensor and were unchanged for the remainder of the experiment.

C3. Figure 4, Panel A is with linear scale, Panel B-D is with logarithmic scale. Why the authors want to have two different scales?

C3. Author's response Figure 4a uses a linear scale to compare the uncalibrated median NO2 EC sensor to the co-located reference NO2 measurement. The NO2 sensor signal differed from the reference measurement sufficiently to allow this to be on a linear scale and to show the reader that the NO2 sensor was able to detect the general trend of the NO2 concentration patterns, but that there was still a large amount of discrepancy between the two measurements.

However, plotting Fig. 4 b to d) on logarithmic axis shows the fit of the calibrated median NO2 sensor with the reference measurement. The improvement of the NO2 measurement across the deployment means that it was difficult to identify times where the concentration estimate contained more error and uncertainty, but the log scale shows this clearly. These higher-error/more uncertain measurements could then be

justified by identifying when other variables exhibited measurements that were outside of their training period ranges. We would therefore like to keep the figure in its current state but are happy to change at the editor's request.

C4. Figure 5 is a nice way to explain the advantage from the ML method. Can the authors do the same for the other ML processing?

C4. Author's response The Boosted Regression Tree gain contributions for each variable was a major reason for using this as a calibration algorithm, and we are glad the reviewer liked Fig. 5. The gain contributions were also analysed for the OX EC BRT algorithm and have been added to the manuscript. This function of the BRT was advantageous as it allows the user to identify the key variables that impact the sensor signals which can then be compared with prior knowledge from laboratory experiments and other studies, thus removing some of the "black box" nature of these algorithms.

C4. Changes to manuscript: Gain contributions from OX BRT added to Figure 5 on page 21.

C4. Author's response Please note that the values used in the pie chart for the NO2 concentration estimate gain contributions have been changed slightly. Whilst adding in the OX pie chart the authors noticed that the previous NO2 plot was an old version, and this has now been changed to the most up-to-date chart. Where cited in the manuscript, values relating to these plots have been updated accordingly.

It is unfortunately not possible to extract the same information from the Gaussian Process implementation that was used in this work. This approach does however provide a prediction uncertainty, see Fig. 4b in manuscript, which is very useful when interpreting the predicted concentrations, in particular when they move into variable space outside of that experienced during the model training dataset.

Linear regression weights for variables can be extracted from the BLR algorithm. However, to make assumptions about the relative importance of each of the sensors to the

algorithm, all the variables, including the reference observations were normalised to between 0 − 1. The BLR analysis was then repeated with the normalised data. This does not change the algorithm and the concentration estimates were identical to those used in the manuscript after the normalisation process.

The resultant weights can be used to indicate that, for the NO2 BLR algorithm, the linear function describing the NO2 sensor measurement contributed the most to the BLR algorithm. Equally, the linear component of the OX sensor measurement was the most important variable for determining the BLR algorithm when predicting the OX concentration estimate. The ability to extract these weights from the BLR analysis is useful for identifying relationships between the sensors, yet this was not included in the manuscript because it overcomplicated the analysis.

These weights output from BLR should not be directly compared to the gain contributions extracted from the BRT, because they are different metrics. The weights from BLR examine the linear relationships between variables whereas the gain contribution from BRT analysed the degree to which each variable contributed to the regression tree decisions – this includes non-linear functions. If the editor wishes, this can be included in the manuscript.

C5. The ML corrected LCS signal still significantly smaller than those measured by the reference instruments especially for the peak values of Ox? Could the authors provide more discussions on this aspect and what could be the possible improvements on LCS or ML.

C5. Author's response Machine learning techniques are very powerful at data interpolation, but often fail when it comes to extrapolation beyond the training data variable space. It is for this reason that linear models can often out perform some ML techniques when using small training datasets. The performance of the ML techniques can be greatly improved by randomly distributing the training data throughout the full time-series in order to cover more variable space. However, this is not a realistic calibration

strategy for low cost sensors and so was not pursued in this work. In Figure 6, at peak [OX] the BRT ML corrected OX concentration estimate does sometimes under-predict the concentration of OX, compared to the reference measurement. This is due to the median OX EC sensor reporting values at these times that are slightly higher than the maximum [OX] observed by the median OX EC sensor during the training period. The inability of the BRT algorithm to extrapolate caused the BRT predicted [OX] estimate to be lower than the reference measurements, in a similar manner to the BRT NO2 prediction. To improve the comparison between the BRT OX concentration estimate and the OX reference measurements more training data is required. This will ensure that the concentration range of [OX] as measured by the EC sensors in the testing period is within the [OX] range in the training data. This was summarised by a few lines which were added to the text on page 9, lines 32 -34.

C5. Changes to manuscript, page 9, lines 32 -34 The ML technique with the lowest RMSE, BRT, bought the OX concentration estimate much closer to the reference observations, see Fig. 6, however, during peaks in OX concentration, the BRT predicted OX concentration estimate was underpredicted due to BRT's inability to extrapolate.

C6. Technical comments: In most cases, the multi-citations were not correctly implemented. For example, page 2 line 8, (Caron et al., 2016),(Jiao et al., 2016) should be (Caron et al., 2016; Jiao et al., 2016). This shall be revised throughout the paper.

C6. Author's response Thanks for notifying the authors about this error, this issue was addressed above in the Reviewer 1 Technical Comment 2.

C7. Figure 3 is not cited in the main text which I assume should appear somewhere in Sect. 3.2.

C7. Author's response Figure 3, showing how increasing the number of EC sensors from 1 to 6 within a cluster improves the agreement between the reference measurement and the median sensor signal, was cited within the manuscript in section 3.1 on Page 5, line 34.

References Friedman, J. H.: Greedy function approximation: a gradient boosting machine, Ann. Statisitcs, 29(5), 1189–1232, 2001.

Roberts, S., Osborne, M., Ebden, M., Reece, S., Gibson, N. and Aigrain, S.: Gaussian processes for time-series modelling., Philos. Trans. A. Math. Phys. Eng. Sci., 371(1984), 20110550, doi:10.1098/rsta.2011.0550, 2013.

Please also note the supplement to this comment:
https://www.atmos-meas-tech-discuss.net/amt-2018-285/amt-2018-285-AC2-supplement.pdf

[Figure]

[Figure]

**Fig. 1.** Bivariate regression – Ordinary Least Squares- was performed on the median NO2 EC sensor (grey) and the NO2 CAPS measurement during the training data set. and the resulting model was applied to the me

a)

Median NO$_2$
(78.89 %)

b)

Median O$_X$
(92.3 %)

Gain contribution for the
BRT NO$_2$ prediction

Gain contribution for the
BRT O$_X$ prediction

**Key:**

■ Median O$_X$ (ppb)                   ■ Median O$_3$ ( Med. O$_X$ – Med. NO$_2$) (ppb)
■ Median NO$_2$ (ppb)                 □ Relative humidity (%)
■ Median CO (ppb)                       ■ Temperature (°C)
■ Median VOC (V)

**Fig. 2.** (Fig. 5 in manuscript): Breakdown of contribution from each variable used by the BRT algorithm to predict the clustered a) NO2 sensor and b) OX concentrations.

[Figure]

| Variable | BLR weight |
|---|---|
| Median VOC sensor | 0.039 |
| RH | -0.087 |
| Temperature | 0.037 |
| Median $NO_2$ EC | 0.73 |
| Median $O_X$ EC | 0.22 |
| Median $O_3$ EC | -0.24 |
| Median CO EC | 0.056 |

$NO_2$ bias: -0.489437

**Fig. 3.** Weights for the BLR-predicted NO2 concentration, with normalised variables prior to analysis.

[Figure]

| Variable | BLR weight |
|---|---|
| Median VOC sensor | 0.11 |
| RH | 0.0055 |
| Temperature | -0.13 |
| Median NO$_2$ EC | -0.33 |
| Median O$_X$ EC | 1.17 |
| Median O$_3$ EC | 0.043 |
| Median CO EC | 0.049 |

O$_X$ bias: -0.484464

**Fig. 4.** Weights for the BLR-predicted OX concentration, with normalised variables prior to analysis.

[Figure]

---

## Author Response (AR2)

**Responding to Reviewer 1 Comments**

**General Comments**

Overall this a well-written, well-organized contribution to the low-cost sensor literature. The authors demonstrate the importance of bespoke sensor calibration using several techniques, including sensor clustering and various statistical and machine learning techniques. Sensor clustering reduces uncertainty due to inter-sensor differences and overall accuracy is improved using several statistical/machine learning techniques.

**Specific Comments**

**C1. How accurate is 'good enough'?**
**I think a bit more context regarding this question would be helpful to readers.**

C1. Author's Response

Thank you to the reviewer for the advice to be clearer when describing the requirements for sensor performance before they are considered able to perform as instruments in the field. The answer to this question is very application dependant and we cannot therefore provide a definitive statement on how good is good enough. It is for this reason that we included the comparison for $NO_2$ measured by two identical reference grade instruments in order to provide some reference for our comparisons. In order to try and expand on this point, we have added some text to the manuscript (page 3, lines 2 - 8) describing the standards set for reference grade instrument performance as set by the EU Directive 2008/50/EC, Annex 1(EU, 2008). Conforming to these standards is an obvious target for low cost sensor measurement performance, but that is not to say that reduced accuracy observations do not hold value providing the uncertainties are quantified.

C1. Changes to manuscript

Page 3, lines 2 - 8

For reference monitors in the UK, $NO_x$, CO and $O_3$ instruments must produce reproducible measurements for three months that are within 5% of the average for a certain concentration in the field, and results that are linear over a set range (EU, 2008). For $NO_x$ this is $0 – 2000$ ppb and $O_3$: 0- 500 ppb and CO: $0 – 50$ ppm to ensure that both rural and urban concentration ranges are taken into account. Although the target performance of low-cost sensors is highly application dependent, these standards do provide a benchmark for comparison and highlight the need not only for high accuracy measurements but also reproducibility over long (months) timescales. In order for low cost sensors to be used in atmospheric monitoring or research applications the uncertainty and reproducibility must be quantified across a range of likely environmental conditions.

**C2. Overall, using SLR and ML techniques seems to be the largest source of improvement. Is sensor clustering even necessary?**

C2. Author's response

Ultimately the clustering and statistical calibration methods are performing different functions in improving sensor performance. In terms of measurement accuracy, the SLR and ML calibration algorithms provide significant improvement over simple linear regression, due to their ability to correct for the multiple cross-sensitivities on sensor signals. In contrast, the function of the clustering approach is not to improve measurement accuracy, but rather sensor reproducibility. As shown in our previous paper (Smith et al., 2017) many sensors show variability in both signal and sensitivity over timescales of days or longer. This variability is very difficult to remove through time averaging, however the lack of correlation of this noise / drift between identical sensors means it can be addressed by instead averaging over multiple sensors. The conclusion of Smith (2017) was that clustering greatly reduces medium-term random noise in the average sensor signal, thus improving confidence in sensor signals and in theory prolonging the time requirements between calibrations.

The time series below illustrates how sensor signals drift apart over time. The plot on the left shows all six sensor signals immediately after calibration to a reference monitor, showing a tight clustering around the median value (red). The plot on the right however shows the drift in individual sensors after a period of 16 days. The use of an average sensor reduces some of this signal variability enabling a more robust calibration to be applied, using algorithms such as SLR or Gaussian Process etc.

[Figure]

**Figure 1. Six individual $O_X$ EC (blue) with the median $O_X$ EC (red), a) immediately after SLR calibration with the reference observations and b) 16 days after the calibration.**

[Figure]

**Figure 2. Six individual NO₂ EC (green) with the median NO₂ EC (purple), a) immediately after SLR calibration with the reference observations and b) 16 days after the calibration.**

The following text was added to the manuscript to emphasise that clustering and ML are used to target different issues. Page 4, line 26 -27.

C2. Changes to manuscript

The clustering approach was used to improve sensor reproducibility as previously discussed in (Smith et al., 2017), whereas the SLR and ML techniques were applied to improve sensor accuracy by correcting for cross sensitivities.

**C3. Does sensor accuracy vary over the observed concentration ranges?**

C3. Author's response

This is an excellent question by the reviewer, and we have performed additional analysis below to investigate this.

For each calibration method used in the paper, the data was 25 % of the observed reference concentration range bins. The Root Mean Squared Error (RMSE) and the Normalised Root Mean Squared Error (NRMSE) were calculated for each concentration

bin and the results for NO2 and OX are summarised in the tables below. The NRMSE was calculated by dividing the RMSE between the reference observations and the sensor values by the mean reference concentration for the respective bin.

Table 1 and 2 nicely displayed how the different analytical techniques improved the sensor performance at different concentrations. Therefore, we decided to include Table 1 (page16) and a description of the results in the manuscript summarising the NO2 RMSE and NRMSEs. The OX summary was very similar so wasn't included, but the authors are happy to include it if the editor wishes.

C3. Changes to manuscript

**Table 1 The NRMSE and RMSE between the NO$_2$ reference and sensor data sets at different concentrations ranges. For each calibration method used in the paper, the data was binned into 25% of the observed reference concentration. The Root Mean Squared Error (RMSE) and the Normalised Root Mean Squared Error (NRMSE) were calculated for each concentration bin and the results for NO$_2$ and O$_X$ are summarised in the tables below. The NRMSE was calculated by dividing the RMSE between the reference observations and the sensor values by the mean reference concentration for the respective bin.**

| Concentration range as a % of the max. conc. of reference NO$_2$ | NRMSE of Reference vs. NO$_2$ concentration estimate (RMSE / ppb) | | | | |
|---|---|---|---|---|---|
| | Median | SLR | BLR | BRT | GP |
| 0 - 25 % | 1.04 (20.7) | 0.59 (11.7) | 0.32 (6.3) | 0.28 (5.6) | 0.29 (5.8) |
| 25 - 50 % | 0.69 (47.5) | 0.19 (13.3) | 0.12 (8.2) | 0.22 (15.2) | 0.11 (7.9) |
| 50 - 75 % | 0.72 (94.9) | 0.23 (30.8) | 0.26 (34.6) | 0.55 (72.5) | 0.26 (33.5) |
| 75 - 100 % | 0.85 (153.1) | 0.10 (17.4) | 0.10 (18.8) | 0.67 (120.0) | 0.10 (18.2) |

Page 9, line 18 - 26

The RMSE and NRMSE was calculated after the application of SLR and ML for different reference concentration ranges to indicate where the greatest improvement of the sensor data occurred (see Table 2). The RMSE and NRMSE (calculated by dividing the RMSE by the mean of the concentration bin) were determined between the reference NO$_2$ observations and the sensor values for four equally spaced reference concentration bins. The ML techniques produced the greatest improvements in the concentration estimates for the lower concentrations of the target measurand where the effect of cross interferences is more significant. The BRT and GP in particular displayed large improvements for the lower NO$_2$ reference observations. At the higher concentrations of NO$_2$, the ML algorithms displayed less improvement, where the conditions were outside those of the training data variable space. This was very noticeable for the BRT algorithm due to its inability to extrapolate.

C3. Author's response

**Table 2. The NRMSE (and RMSE) between the $O_X$ reference and sensor data sets at different concentration ranges.**

| Concentration range as a % of the max. conc. of reference $O_X$ | NRMSE of Reference vs. $O_X$ concentration estimate (RMSE / ppb) | | | | |
|---|---|---|---|---|---|
| | Median | SLR | BLR | BRT | GP |
| 0 - 25 % | 0.21 (11.0) | 0.16 (8.4) | 0.10 (5.4) | 0.12 (6.0) | 0.18 (9.2) |
| 25 - 50 % | 0.30 (26.4) | 0.12 (10.2) | 0.11 (9.4) | 0.11 (9.7) | 0.14 (12.4) |
| 50 - 75 % | 0.36 (50.4) | 0.12 (16.3) | 0.12 (16.1) | 0.10 (14.0) | 0.16 (22.4) |
| 75 - 100 % | 0.52 (116.1) | 0.20 (44.7) | 0.26 (58.0) | 0.49 (110.9) | 0.27 (60.6) |

C3. Changes to manuscript

Page 10 line 8 - 11.

The NRMSE was calculated for 4 equally sized reference $O_X$ concentration bins for each analytical method used, in a similar manner to Table 2 for $NO_2$. The NRMSE improved for SLR and the ML algorithms across all concentration ranges, with BLR

10 and BRT optimal for reducing the error estimate the most. The error was the highest at the higher $O_X$ concentrations for BRT, which was expected due to BRTs inability to extrapolate.

**C4. Technical Comments p1 l30. 'site'->situated**

C4. Authors response

15 The wording has been changed on page 1, line 32.

**C5 . p2 l8, l19, l21... Check reference parentheses throughout p4 l25.**

C5. Author's response

Removed the extra brackets between multiple references and inserted a semi-colon to differentiate two citations for the same

20 reference.

**C6. Also Hagan et al. AMT 2018**

C6. Author's response

Added the reference into the manuscript on page 5, line 3 as it was relevant to the manuscript.

References

EU: Directive 2008/50/EC of the European Parliament and of the Council of 21 May 2008 on ambient air quality and cleaner

air for Europe, Eur. Union, 1–62, 2008.

Smith, K., Edwards, P. M., Evans, M. J. J., Lee, J. D., Shaw, M. D., Squires, F., Wilde, S. and Lewis, A. C.: Clustering approaches that improve the reproducibility of low-cost air pollution sensors, Faraday Discuss., 00(0), 1–17,
5   doi:10.1039/C7FD00020K, 2017.

**Responding to Reviewer 2 Comments**

**General Comments**

10   Low cost sensors (LCS) playing an emerging role in the urban environmental monitoring with respect to the possibility of setting up a densely populated gridded network. Nevertheless, the detection limit, the stability and the real-time calibration were in general of question or with difficulty to overcome. In this study, the authors try to use the machine learning (ML) method to enhance of the data quality of LCS which is in general fit the effort of the community to improve the data quality of LCS. The paper is within the scope of AMT and I have the following comments for the authors to consider before
15   publication.

**C1. The machine learning method is used to improve the data quality of the LCS. The improvement is clear but still without in-depth explanations. The scientific paper shall not be looks like simply magic. I will be convinced if the authors can provide much more examples as the authors also wrote in their conclusions. Moreover, I did see much**
20   **better comparison results from LCS (the Cambridge group for the same campaign) with the CAPS instrument on NO2 and other parameters like $O_3$, CO, etc. So, I wonder if the results presented in this paper can be improved further.**

C1. Author's response

We thank the reviewer for this comment, as this is something we really wanted to avoid and have therefore added more detail
25   in order to try and be more explicit about this. During the analysis section of this work the authors made sure that the ML techniques used provided outputs on the decisions they made that could then be compared with laboratory experiments and previous sensor studies, in order to make the methods used not seem like black boxes.

This was underpinned by our choice of ML techniques; BRT was chosen because of the function to extract out the variables gain contributions, GP could produce the uncertainty for each predicted data point and the weights associated with each
30   variable can be extracted from the BLR algorithm (see C4 with Figs. 4 and 5 of this document). The manuscript aimed to compare results from using different techniques on the same dataset and therefore, fully comprehensive explanations of different ML techniques is beyond the scope of this paper. However, we recognise that this was not made clear enough in the manuscript so have added some more detail in the text and some more citations to detailed descriptions of the techniques.

With respect, the authors have not seen any publications from the Cambridge group on this and cannot comment on unpublished work.

Whilst there are many references to studies where LCS have been used successfully in the field, the scope of this manuscript relates to the improvement of low-cost sensor performance for deployment over longer periods of time and possible calibration

5   strategies that would enable this.

C1. Changes to manuscript

Gaussian process reference inserted:

10   Gaussian process for time series modelling, S. Roberts, M. Osborne, M. Ebden, S. Reece, N. Gibson and S. Aigrain (Roberts et al., 2013)

Page 7, line 27.

XGBoost reference inserted:

15   Greedy function approximation: a gradient boosting machine (Friedman, 2001).

Page 7, line 17.

**C2. Sect. 3.2: during the training period, what kind of regression method is used to calibrate the sensors? According to Cantrell, 2008 (Cantrell, C. A.: Technical Note: Review of methods for linear least-squares fitting of data and**

20   **application to atmospheric chemistry problems, Atmos. Chem. Phys., 8, 5477–5487, 2008.), bivariate regression algorithm is required to retrieve the robust slope.**

C2. Author's response

Thanks for bringing to our attention that the SLR method described in Section 3.2 was unclear. More text has been added to

25   better describe the linear regression process. There were four different types of analytical techniques used in turn to examine the performance of ML versus simple linear regression (SLR). In section 3.2, SLR was used to calibrate the EC sensors against their respective reference instruments.

Using $NO_2$ as an example, linear parameters in the form of $y = mx + c$ were determined using a linear least squares fit between

30   the $NO_2$ CAPS reference instrument and the median $NO_2$ EC sensor. This linear relationship was calculated for the first five days of the deployment – the same five days that were used as the training period for the ML analysis.

Text has been added on Page 6 lines 6 – 10, to provide further detail about SLR.

The training period for the NO$_2$ EC sensors and NO$_2$ reference measurements was also re-analysed using bivariate regression (Ordinary Least Squares) and the resulting model was applied to the median NO$_2$ sensors over the testing period. This produced a bivariate regression NO$_2$ prediction, shown in green in Figure 1. The regression was performed using the Python statsmodels package.

[Figure]

**Figure 1. Bivariate regression – Ordinary Least Squares- was performed on the median NO$_2$ EC sensor (grey) and the NO$_2$ CAPS measurement during the training data set. and the resulting model was applied to the median NO$_2$ to produce a bivariate regression (BR) predicted trace (green).**

10 The RMSE was calculated between the BR NO$_2$ prediction and the NO$_2$ reference measurements in the testing period and was found to be 14.6 ppb. The bivariate regression prediction therefore contains more error than the simple linear regression in the manuscript (10.42 ppb). This does not change the outcome of the paper which aims to use ML to improve the quality of the NO$_2$ sensors by correcting for cross interferences and therefore has not been added to the manuscript. The SLR was used to show the calibration of the sensors using linear regression and essentially set a baseline for the improvements.

C2. Changes to manuscript

Using the NO$_2$ EC as an example, linear parameters in the form of $y = mx + c$ were determined using a linear least squares fit between the NO$_2$ CAPS reference instrument and the median NO$_2$ EC sensor for the first five days of the sensor instrument

deployment. Once trained in this manner, these linear calibration factors based on SLR were used to calibrate the median $NO_2$ sensor and were unchanged for the remainder of the experiment.

**C3. Figure 4, Panel A is with linear scale, Panel B-D is with logarithmic scale. Why the authors want to have two different scales?**

C3. Author's response

Figure 4a uses a linear scale to compare the uncalibrated median $NO_2$ EC sensor to the co-located reference $NO_2$ measurement. The $NO_2$ sensor signal differed from the reference measurement sufficiently to allow this to be on a linear scale and to show the reader that the $NO_2$ sensor was able to detect the general trend of the $NO_2$ concentration patterns, but that there was still a large amount of discrepancy between the two measurements.

However, plotting Fig. 4 b to d) on logarithmic axis shows the fit of the calibrated median $NO_2$ sensor with the reference measurement. The improvement of the $NO_2$ measurement across the deployment means that it was difficult to identify times where the concentration estimate contained more error and uncertainty, but the log scale shows this clearly. These higher-error/more uncertain measurements could then be justified by identifying when other variables exhibited measurements that were outside of their training period ranges.
We would therefore like to keep the figure in its current state but are happy to change at the editor's request.

**C4. Figure 5 is a nice way to explain the advantage from the ML method. Can the authors do the same for the other ML processing?**

C4. Author's response

The Boosted Regression Tree gain contributions for each variable was a major reason for using this as a calibration algorithm, and we are glad the reviewer liked Fig. 5. The gain contributions were also analysed for the $O_X$ EC BRT algorithm and have been added to the manuscript. This function of the BRT was advantageous as it allows the user to identify the key variables that impact the sensor signals which can then be compared with prior knowledge from laboratory experiments and other studies, thus removing some of the "black box" nature of these algorithms.

C4. Changes to manuscript: Gain contributions from $O_X$ BRT added to Figure 5 on page 21.

[Figure]

**Figure 5: Breakdown of contribution from each variable used by the BRT algorithm to predict the clustered a) NO₂ sensor and b) Oₓ concentrations.**

C4. Author's response

Please note that the values used in the pie chart for the $NO_2$ concentration estimate gain contributions have been changed slightly. Whilst adding in the $O_X$ pie chart the authors noticed that the previous $NO_2$ plot was an old version, and this has now been changed to the most up-to-date chart. Where cited in the manuscript, values relating to these plots have been updated accordingly.

It is unfortunately not possible to extract the same information from the Gaussian Process implementation that was used in this work. This approach does however provide a prediction uncertainty, see Fig. 4b, which is very useful when interpreting the predicted concentrations, in particular when they move into variable space outside of that experienced during the model training dataset.

Linear regression weights for variables can be extracted from the BLR algorithm. However, to make assumptions about the relative importance of each of the sensors to the algorithm, all the variables, including the reference observations were normalised to between 0 – 1. The BLR analysis was then repeated with the normalised data. This does not change the algorithm and the concentration estimates were identical to those used in the manuscript after the normalisation process.

[Figure]

| Variable | BLR weight |
| --- | --- |
| Median VOC sensor | 0.039 |
| RH | -0.087 |
| Temperature | 0.037 |
| Median $NO_2$ EC | 0.73 |
| Median $O_X$ EC | 0.22 |
| Median $O_3$ EC | -0.24 |
| Median CO EC | 0.056 |

$NO_2$ bias: -0.489437

**Figure 3. Weights for the BLR-predicted NO₂ concentration, with normalised variables prior to analysis.**

[Figure]

| Variable | BLR weight |
|---|---|
| Median VOC sensor | 0.11 |
| RH | 0.0055 |
| Temperature | -0.13 |
| Median $NO_2$ EC | -0.33 |
| Median $O_X$ EC | 1.17 |
| Median $O_3$ EC | 0.043 |
| Median CO EC | 0.049 |

$O_X$ bias: -0.484464

**Figure 4. Weights for the BLR-predicted $O_X$ concentration, with normalised variables prior to analysis.**

The resultant weights can be used to indicate that, for the $NO_2$ BLR algorithm, the linear function describing the $NO_2$ sensor measurement contributed the most to the BLR algorithm. Equally, the linear component of the $O_X$ sensor measurement was

5 the most important variable for determining the BLR algorithm when predicting the $O_X$ concentration estimate. The ability to extract these weights from the BLR analysis is useful for identifying relationships between the sensors, yet this was not included in the manuscript because it overcomplicated the analysis.

These weights output from BLR should not be directly compared to the gain contributions extracted from the BRT, because they are different metrics. The weights from BLR examine the linear relationships between variables whereas the gain

10 contribution from BRT analysed the degree to which each variable contributed to the regression tree decisions – this includes non-linear functions.

If the editor wishes, this can be included in the manuscript.

**C5. The ML corrected LCS signal still significantly smaller than those measured by the reference instruments especially**

15 **for the peak values of Ox? Could the authors provide more discussions on this aspect and what could be the possible improvements on LCS or ML.**

C5. Author's response

Machine learning techniques are very powerful at data interpolation, but often fail when it comes to extrapolation beyond the

20 training data variable space. It is for this reason that linear models can often out perform some ML techniques when using small training datasets. The performance of the ML techniques can be greatly improved by randomly distributing the training

data throughout the full timeseries in order to cover more variable space. However, this is not a realistic calibration strategy for low cost sensors and so was not pursued in this work.

In Figure 6, at peak [$O_X$] the BRT ML corrected $O_X$ concentration estimate does sometimes under-predict the concentration of $O_X$, compared to the reference measurement. This is due to the median $O_X$ EC sensor reporting values at these times that are slightly higher than the maximum [$O_X$] observed by the median $O_X$ EC sensor during the training period. The inability of the BRT algorithm to extrapolate caused the BRT predicted [$O_X$] estimate to be lower than the reference measurements, in a similar manner to the BRT $NO_2$ prediction. To improve the comparison between the BRT $O_X$ concentration estimate and the $O_X$ reference measurements more training data is required. This will ensure that the concentration range of [$O_X$] as measured by the EC sensors in the testing period is within the [$O_X$] range in the training data. This was summarised by a few lines which were added to the text on page 9, lines 32 -34.

C5. Changes to manuscript, page 9, lines 32 -34

The ML technique with the lowest RMSE, BRT, bought the $O_X$ concentration estimate much closer to the reference observations, see Fig. 6, however, during peaks in $O_X$ concentration, the BRT predicted $O_X$ concentration estimate was underpredicted due to BRT's inability to extrapolate.

**C6. Technical comments**: **In most cases, the multi-citations were not correctly implemented. For example, page 2 line 8, (Caron et al., 2016),(Jiao et al., 2016) should be (Caron et al., 2016; Jiao et al., 2016). This shall be revised throughout the paper.**

C6. Author's response

Thanks for notifying the authors about this error, this issue was addressed above in the Reviewer 1 Technical Comment 2.

**C7. Figure 3 is not cited in the main text which I assume should appear somewhere in Sect. 3.2.**

C7. Author's response

Figure 3, showing how increasing the number of EC sensors from 1 to 6 within a cluster improves the agreement between the reference measurement and the median sensor signal, was cited within the manuscript in section 3.1 on page 5, lines 34.

**References**

[revised manuscript text omitted]